# Time-Course Changes of Extracellular Matrix Encoding Genes Expression Level in the Spinal Cord Following Contusion Injury—A Data-Driven Approach

**DOI:** 10.3390/ijms22041744

**Published:** 2021-02-09

**Authors:** Andrea Bighinati, Zahra Khalajzeyqami, Vito Antonio Baldassarro, Luca Lorenzini, Maura Cescatti, Marzia Moretti, Luciana Giardino, Laura Calzà

**Affiliations:** 1Department of Veterinary Medical Science, University of Bologna, Ozzano dell’Emilia, 40064 Bologna, Italy; andrea.bighinati@unibo.it (A.B.); luca.lorenzini8@unibo.it (L.L.); luciana.giardino@unibo.it (L.G.); 2Fondazione IRET, Ozzano dell’Emilia, 40064 Bologna, Italy; zahra.khalajzeyqami2@unibo.it (Z.K.); maura.cescatti@unibo.it (M.C.); marzia.moretti3@unibo.it (M.M.); 3Interdepartmental Center for Industrial Research in Life Sciences and Technologies, University of Bologna, Ozzano dell’Emilia, 40064 Bologna, Italy; vito.baldassarro2@unibo.it; 4Department of Pharmacy and Biotechnology, University of Bologna, 40126 Bologna, Italy; 5Montecatone Rehabilitation Institute, 40026 Imola (BO), Italy

**Keywords:** spinal cord injury, extracellular matrix, secondary degeneration, remyelination, timp1, inflammation

## Abstract

The involvement of the extracellular matrix (ECM) in lesion evolution and functional outcome is well recognized in spinal cord injury. Most attention has been dedicated to the “core” area of the lesion and scar formation, while only scattered reports consider ECM modification based on the temporal evolution and the segments adjacent to the lesion. In this study, we investigated the expression profile of 100 genes encoding for ECM proteins at 1, 8 and 45 days post-injury, in the spinal cord segments rostral and caudal to the lesion and in the scar segment, in a rat model. During both the active lesion phases and the lesion stabilization, we observed an asymmetric gene expression induced by the injury, with a higher regulation in the rostral segment of genes involved in ECM remodeling, adhesion and cell migration. Using bioinformatic approaches, the metalloproteases inhibitor *Timp1* and the hyaluronan receptor *Cd44* emerged as the hub genes at all post-lesion times. Results from the bioinformatic gene expression analysis were then confirmed at protein level by tissue analysis and by cell culture using primary astrocytes. These results indicated that ECM regulation also takes place outside of the lesion area in spinal cord injury.

## 1. Introduction

Spinal cord injury (SCI) is a devastating, incurable condition which can severely limit motor abilities but also causes a number of sensory, visceral and systemic symptoms and complications. According to the Global Burden of Diseases, Injuries and Risk Factors Study (GBD) 2019, the age-standardized incidence rate for SCI is 13 per 100,000, while the prevalence is 368 per 100,000 and 90% of cases are due to falls or accidents [1].

When a traumatic injury occurs, a cascade of cellular and molecular events follows the mechanical impact, propagating the primary damage and causing it to spread during a phenomenon known as “secondary degeneration.” Secondary degeneration has been described in both animal models and humans [2] and includes an acute phase beginning immediately after SCI, involving vascular damage, excitotoxicity, free radical formation, inflammation, edema and necrotic cell death. This is followed by the subacute phase, characterized by apoptosis, demyelination, axonal die-back and Wallerian degeneration, matrix remodeling and scar formation. The final component is the chronic phase, involving the formation of a cystic cavity (syringomyelia, which is present in humans and to a lesser extent in rodents [3], progressive axonal die-back and evolution toa fibrotic scar. The secondary lesion evolves differently rostrally and caudally to the lesion; syringomyelia extension being more severe caudally to the lesion [4], whereas white matter lesion appears to be more severe rostrally to the lesion epicenter [5].

In this complex context, various roles have been attributed to the extracellular matrix (ECM) components over the past decades with regard to the determination of SCI evolution, including free growth factor availability, axonal regeneration, resident and incoming inflammatory cell activation, cell proliferation and migration and attempts at myelin repair. Altered ECM composition after injury is caused by the degradation of certain ECM components and the increased synthesis and activity of others [6]. Different metalloproteases (MMPs), for example, can be released by activated microglia/macrophages, a mechanism triggered by hyaluronan fragments, tenascins and sulfated proteoglycans liberated in the site of injury [7]. The inhibitory role played by the robust up-regulation of chondroitin sulphate proteoglycans on axonal regrowth following injury has been recognized for many years [8]. In the event of injury to the meninges, incoming fibroblasts produce structural components of the ECM, such as fibronectin, collagen and laminin [2]. ECM composition changes are one of the major mechanisms which limit axonal sprouting and regrowth [9] and which alter the process of oligodendrocyte precursor cell (OPC) maturation into newly myelinating oligodendrocytes, responsible for remyelination attempts in the white matter after lesion [10] and regarded as a potential therapeutic target [11].

A clear picture of the cascade of changes in ECM composition and structure at different times after trauma, however, is still elusive, due in part to the complex and evolving nature of this lesion, with no clear distinction between causal and secondary molecular events [12]. Moreover, the succession of partly overlapping pathophysiological events (mechanical trauma, inflammation, excitotoxicity, demyelination and remyelination attempts, neurodegeneration via necrosis, apoptosis, Wallerian and retrograde degeneration, scar formation, etc.) creates a number of confounding interactions. All of these unpredictable variables challenge the classical hypothesis-oriented study design, while the burgeoning availability of “omic” techniques offers a unique opportunity for a target discovery strategy, one which combines a robust experimental design with the bioinformatic analysis of experimental data and confirmatory in vivo and in vitro experiments.

We therefore designed a study to investigate the ECM protein encoding genes in the spinal cord areas immediately rostral and caudal to the lesion core, using an mRNA array profile and Real Time-Polymerase Chain Reaction (RT-PCR) comprising around 100 genes, with three timepoints corresponding to the inflammatory phase (24 h post-lesion), ongoing secondary degeneration (8 days post-lesion) and the chronic phase (45 days post-lesion), focusing our attention on the segments where most of the ascending and descending tracts were compromised by the lesion. The hypothesis generated from the bioinformatic analysis of the gene expression results was then validated at protein level and by in vitro experiments.

## 2. Results

### 2.1. Animals and Lesion Characterization

The spinal cord area directly affected by the mechanical lesion is illustrated in a schematic coronal section of the lesion segment of the spinal cord, including the area directly impacted by the punch (Figure 1A, gray area). The general wellness of the animals was monitored by body weight (Figure 1B) and an initial decrease was observed, followed by a recovery starting from 8 Days Post Lesion (DPL). All animals showed a mild to moderate locomotor impairment following SCI, as assessed by the Basso, Beattie, Bresnahan (BBB) locomotor scale and a spontaneous recovery beginning at 7 DPL, which stabilized at 14 DPL (Figure 1C). None of the study rats died. 

Lesion extension was analyzed by histology, using both longitudinal and coronal sections. The anatomical schema in Figure 1D shows the general sampling strategy for tissue analysis and includes the area directly affected by the mechanical lesion (center of the lesion) and the rostral and caudal segments affected by secondary degeneration. Panels in Figure 1E show representative micrographs of coronal sections collected at the corresponding rostro-caudal level at 1, 8 and 45 DPL. Sections collected at the center of the lesion at 1 and 8 DPL were almost destroyed and the resulting cavitation is evident at 45 days. Figure 1F shows the rostro-caudal and dorso-ventral extension of the lesion at 45 DPL, as obtained by 3D reconstruction of longitudinal sections from a representative animal. The contusive lesion and secondary degeneration extended to a dorso-ventral depth of approximately 2000 µm, reaching the ventral horn as expected and 6000 µm in rostro-caudal direction, from segment T7 to segment T11.

### 2.2. Gene Expression Profile in the Spinal Cord Segment Rostrally and Caudally to the Lesion Center of the Lesion

The expression of genes encoding for proteins of the ECM was investigated at 1, 8 and 45 DPL following spinal cord lesion, rostrally and caudally to the lesion, therefore in segments which are not directly involved in the mechanical lesion but affected by the secondary degeneration process. This was done by comparing the expression profile of 84 genes to the respective segments of intact animals and the list of the investigated genes is reported in Figure 2. For biological averaging and variance reduction, samples from each group were pooled as for microarray experiments [13,14,15].

The relative gene expression is shown as a heat map showing correlated gene expression across each group and time point (Figure 2). The relative fold change for each gene is shown in the table next to the clustergram, with red being the maximum and green the minimum difference of expression from the median of each gene analyzed. Due to the low expression in all experimental groups (threshold > 35 Ct), we excluded the *Mmp1* gene from further investigation. The fold of difference normalized on the corresponding spinal cord segment in intact animals is reported in the table, where up-regulated genes are shown in red and down-regulated genes are shown in green.

The main focus of the study was to compare the ECM gene expression in the spinal cord segments rostral and caudal to the lesion. The two hierarchical clusterization analyses obtained using Gene Globe software revealed a similar regulation between the two segments soon after the lesion (1 DPL), characterized mainly by the up-regulation of the genes involved in ECM remodeling (*Timp1*, *Mmps*, *Sell*, *CD44*, *Fn1*). However, the rostral segment showed a higher number of up-regulated genes (rostral, *n =* 21; caudal *n =* 6) and a lower number of down-regulated genes (rostral, *n =* 1; caudal *n =* 13), with *CD44*, *Mmp3*, *Sell*, *Thbs2*, *Timp1* and *Tnc* commonly up-regulated in the two segments. At longer time points from the lesion, the caudal segment showed a similar pattern of gene regulation (8 and 45 DPL), with a slight down-regulation of most analyzed genes. The rostral segments also showed a similar pattern of gene regulation compared to the caudal segments, while the rostral segment clustered alone at 45 DPL, indicating that the overall gene expression regulation differed from the other time points. 

At this time point, only the *Timp1* gene was still up-regulated in both segments, while 10 and 5 genes were down-regulated in the rostral and caudal segments respectively, with *Col8a1* the only common down-regulated gene in both segments. 

To further investigate the similarities and differences between the rostral and caudal segments in response to spinal cord lesion, we extracted the highly-responsive genes (fold of changes > 4) from the PCR array analysis, grouping them as similar (Figure 3A) or different regulation patterns (Figure 3B) between the two segments. In each graph, the circle represents the rostral and the square the caudal segment, while the dotted horizontal line indicates the intact spinal cord.

The highly responsive genes showing the same regulation pattern between the two segments are mostly involved in ECM reorganization, with a strong representation of Mmps and related proteins (*Mmp3*, *Mmp8*, *Mmp12*, *Timp1*, *CD44*, *Sell* and *Tnc*), while the rostral segment responds to injury with an overexpression of collagen genes (*Col1a1*, *Col3a1* and *Col8a1*) and the related *Postn*, with an up-regulation in early phases (1 and 8 DPL) and a down-regulation at 45 DPL. Moreover, the *Thbs1* gene showed the same temporal pattern, while decreased expression in the caudal segment and the Tgfβ induced gene (*Tgfbi*) were strongly down-regulated in the rostral segment at 45 DPL, while being up-regulated in the caudal segment at the same time point.

Since we used a ready-made PCR array profile, we included the mRNA expression analysis of 14 additional genes encoding for proteins of the neural ECM performed by RT-qPCR, also finding different mRNA expression levels between the two segments of spinal cord for these genes over the time points analyzed. We first confirmed the data obtained using PCR arrays for *CD44*, *Fn1*, *Postn*, *Sell* and *Tnc*, showing a strong up-regulation in both segments at 1 and 8 DPL (Appendix A). Among the other genes, we found a different regulation in the two segments in 3 genes (*Agrin*, *Bcan* and *Slit2*) in the acute phase (1 DPL) and 1gene (*Lgals1*) in the sub-acute phase (8 DPL) (Figure 4A). Agrin is involved in the regulation of neurite outgrowth and synapse formation; *Bcan* encodes for brevican, a chondroitin sulfate proteoglycan; Slit2 regulates axonal guidance in the spinal cord, while *Lgals1*, which encodes for Galectin 1, a beta-galactoside binding protein involved in cell-cell and cell-matrix interaction, showed a statistically significant up-regulation in the rostral segment compared to the caudal segment at 8 DPL (2.5 vs. 1.5 Log2 fold change). No significant variations either in the rostral and caudal segments were observed for *Cspg4*, *Slit1*, *Tnr* or *Ntn1* (Figure 4B).

Moreover, we performed a correlation between the gene expression levels from single animals and the related BBB score, finding no differences for all the analyzed genes (Appendix A). 

### 2.3. Bioinformatic Analysis and PPI Interaction from Cluster Analysis

We then performed bioinformatic analysis of all genes in the array and RT-PCR single gene analysis. We initially classified the up- and down-regulated genes for each segment and time point using the Panther bioinformatic platform for gene enrichment (Appendix A). Gene classification was based on their biological process, as defined by Gene Ontology (GO) analysis. We confirmed the difference observed with the PCR Array clustergram analysis and identified the common biological processes at 1 DPL in both rostral and caudal segments, especially for up-regulated genes. The principal classes of biological process showing different regulation modes between the segments on either side of the lesion epicenter were biological adhesion (GO: 0022610) and cellular process (GO: 0009987) but we also found alterations in immune system process (GO: 0002376), localization (GO: 0051179), biological regulation (GO: 0065007) and developmental process (GO: 0032502).

To identify the major genes involved in ECM remodeling following SCI, the list of genes profiled by the Extracellular Matrix & Adhesion Molecules PCR Array from Qiagen, with the addition of RT-PCR single cell analysis, was analyzed for the protein-protein interaction (PPI) via the String online database and the information used to visualize the ECM cluster on Cytoscape (v3.7.2). Subsequently, the gene expression profiles obtained via PCR array and RT-qPCR for each region and time point were superimposed on to the ECM cluster to include the genes with at least 2-fold change variations. The new sub clusters were filtered by node degree to define a maximum of 10 hub genes for every condition and the results are shown in Figure 5, where the color intensity indicates the folds of gene expression regulation (red, up-regulation; green, down-regulation) and circle size indicates node degree. At 1 DPL, we observed that *Timp1* was the most up-regulated gene in both the rostral and caudal segments, with *Cd44* as the most relevant nodes in both segments, while *Fn1* seemed to be a relevant node in the rostral but not the caudal segment. *Timp1* also maintained a central role at 8 and 45 DPL in both segments, although its expression progressively declined. Notably, *Timp1* was still slightly up-regulated in the stabilized chronic lesion (45 DPL), while all other altered genes (collagens and laminins) were down-regulated. 

To validate these bioinformatic results, we performed the Timp1 protein analysis in the spinal cord tissue (rostral segments) at the same time points and the results are shown in Figure 6A,B, confirming a slight increase and then a progressive Timp1 content decline over the experimental time frame.

### 2.4. In Vitro Evaluation of the Cd44 and Timp1 Response to Inflammation and Timp1 Protein Quantification in the Spinal Cord

To confirm data from mRNA expression, we quantified Timp1 protein level in the segment rostral to the lesion at 1, 7 and 60 DPL (Figure 6A,B). Quantification results were normalized at each time point on the Timp1 protein level in the same segment of spinal cord from healthy animals, used as control. To minimize inter-gels variability, unlesioned animals were distributed in each gel and inter-gel normalization was made both versus b-actin (housekeeping protein) and lesioned versus unlesioned animals. Timp1 protein at 1 DPL was slightly but not significantly up-regulated in lesioned animals. On the contrary, a significant downregulation of the Timp1 protein level was observed at 7 and 60 days compared to unlesioned animals.

To demonstrate the direct link between inflammation and overexpression of the two genes identified in vivo, we used primary culture of astrocytes (Figure 6C), one of the main responsible for Cd44 and Timp1 production, exposed to a cytokine mix containing molecules up-regulated in the CSF 24 h after SCI [16] able to block remyelination [17,18]. Gene expression results are shown in Figure 6D,E. Both *Cd44* and *Timp1* (isoform b) were up-regulated by exposure to the cytokines (Student’s *t*-test; *Cd44*, *p =* 0.0006; *Timp1*, *p =* 0.0118). Interestingly, we also analyzed the isoform “a” (precursor) of the *Timp1* gene, which was not regulated by cytokine exposure (data not shown). Single ΔCt data for each analyzed gene are shown in Appendix A.

## 3. Discussion

Spinal cord trauma is followed by an extremely complex biological syndrome involving different pathophysiological processes (hemorrhage, inflammation, excitotoxicity, demyelination, axonal degeneration, cell death, scar formation, etc.) which evolve over months and even years in humans [2]. Efforts to find a molecular signature which is also able to drive drug discovery is complicated by the impressive regulation of dozens of different cell types and hundreds of molecules which occurs after injury [19] and by the highly personalized nature of each lesion, making each patient’s medical history unique.

All components of the neural ECM (interstitial, perineuronal, basement membrane) are recognized as players in SCI evolution, repair attempts and therefore in functional outcome in chronic lesions [12]. Most published reports, however, describe ECM variations at the core of the lesion, focusing on scar formation [20] and to the best of our knowledge, no data is available on the surrounding segments, where the ascending (caudal segment) and descending (rostral segment) nerve pathways reside and where a bridge between the pathological microenvironment of the scar and intact tissue is likely to take place [5,12]. In these areas, active neurons, reactive glia, including astrocytes, microglia and NG2-OPCs are present and play a possible role in ECM modification [21].

To demonstrate the molecular regulation of this microenvironment, spared from mechanical injury and scarring on one hand but hit by a veritable “storm” of pathological changes on the other, in this study we explored the transcriptome profile of the ECM-encoding genes in the spinal cord tissue adjacent to the injury site. Analysis were focused on the protein component of the ECM through gene expression modulation and does not consider the hyaluronan component. We examined both the spatial and temporal changes, examining the spinal cord segments adjacent to the lesion center of the lesion, in both rostral and caudal directions and analyzing three time points after lesion corresponding to the direct effects of the mechanical impact (24 h post-lesion; 1 DPL), secondary degeneration (8 DPL) and the chronic, stabilized phase (45 DPL).

First of all, we carefully characterized the lesion extension at the investigated times. The areas directly affected by the Impactor tip included the gray matter (dorsal and part of the ventral horn) and the white matter with regard to the dorsal funiculus, which includes the ascending fasciculus gracilis and the descending corticospinal tracts. Following contusion injury, the lesion developed a cystic cavity at the center of the lesion, which spread coronally and longitudinally in both rostral and caudal directions from the 1.5 mm wide impact, extending for more than 4.0 mm across the center of the lesion at 3 DPL, then expanding further, surrounded by scar tissue, an evolution which corresponds to the literature in the field [4,22,23,24]. Quantitative MRI indicated that most of the axons directly affected by the impact degenerated rostrally to the injury site, while minor damage was observed caudally [25]. Axonal damage and myelin pathology in the descending corticospinal tract were also clear, both caudally and rostrally to the injury, due to axonal retraction (rostral) and axonal dieback (caudal), expanding for up to 8 weeks even at a long distance from the injury site.

Overall, we observed an unexpected asymmetry in ECM gene expression regulation in the rostral compared to the caudal segment. Grouping the results of the gene array analysis in “active lesion phases” including 1 and 8 DPL and considering ≥ 2 folds regulation as significant, we observed that 28 out of 80 genes were up-regulated in the rostral segment, while only 10 were up-regulated in the caudal segment. On the contrary, 20 genes were down-regulated in the caudal segment, while none were down-regulated in the rostral segment. It should also be noted that the more strongly up-regulated genes, such as *CD44*, *Mmp3*, *Sell*, *Timp1*, *Tnc*, were regulated in both segments. At lesion stabilization (45 DPL), 10 genes were down-regulated and only one up-regulated (*Timp1*) in the rostral segment, while 3 were down-regulated and one up-regulated (*Timp1*) in the caudal segment. Notably, it has been described that other proteins not related to the ECM are differentially regulated at the center of the lesion and in the rostral and caudal segments [26,27,28], thus confirming the asymmetric profile of molecular regulation around the injury site. Since the meninges were undamaged in the contusive model, these molecular changes do not involve peripheral cells such as fibroblasts but are described as mainly due to astrocytes, pericytes and other glial cell types [29].

Even if it has been described that Mmp1 protein is strongly activated after SCI in both humans and animals [30,31], this protein activation is not confirmed at gene expression level in our model. However, Mmp1 is normally in an inactive state as a pro-proteinase requiring the proteolysis from other metalloproteinases, like Mmp3, for its activation [32] (https://reactome.org/content/detail/R-RNO-1592297). In fact, we found a strong upregulation of Mmp3 at 1 DPL which is sustained also at 8 DPL. Thus, we hypothesized that the absence of Mmp1 expression is not correlated to the absence of this proteinase in the lesioned spinal cord but rather to the activation of Mmp1 at protein level.

With regard to highly regulated genes (×4 fold change), different collagen and collagen-related genes (*Col1a1*, *Col3a1*, *Col8a1*, *Postn*) were strongly responsive in the rostral but not in the caudal segment. Type 1 collagen has already been described as up-regulated in SCI models in the segment rostral to the lesion [33] but a difference between the rostral and caudal segments has never been described. In the healthy CNS, fibril-forming collagens are present in the vessel structure and in meningeal membranes, while non-fibril-forming collagen proteins are widely distributed throughout the CNS, where they play a major role in CNS wiring and repair [34,35]. Although collagen IV is the most widely studied component of the fibrous scar tissue which develops after CNS injury [36], an up-regulation of collagen 1is also seen as an activator of astrocytes, driving them to a scar-forming phenotype [37]. Interestingly, Col1a1-positive cells have recently been identified as perivascular stromal cells activated following brain and spinal cord injuries, participating in fibrosis scar formation and secreting retinoic acid and other fundamental signaling molecules [38]. The *Thbs1* gene, which encodes for thrombospondin 1, is a key astrocyte-derived factor regulating synaptogenesis in the developing brain [39], and is reactivated in case of injuries and inflammation [40,41]. Thbs1 has been also described as a synaptogenic molecule expressed by astrocytes in the perineuronal process and involved in the recovery of excitatory synapses on axotomized motor neurons in the CNS [42]. Moreover, together with Tgfβ, thrombospondin 1 levels rapidly increase at the injury site and both proteins are implicated in angiogenesis, scar deposition and inflammation, as well as affecting astrocyte mobility [43]. Finally, Tgfbi, a collagen-related protein, seems to be involved in both axonal regeneration [44] and scar formation [45] in the spinal cord following injury. These collagen-related genes are not regulated in the caudal segment. 

Several genes were also equally regulated in the rostral and caudal segments in the acute phase of the lesion (1 DPL) and most were then down-regulated at 45 DPL. The mostly up-regulated genes included *Cd44*, *Mmp12*, *Mmp3*, *Mmp8*, *Timp1*, *Sell* and *Tnc*, indicating the Mmps-related mechanisms as those most affected by injury. Using a bioinformatic approach, we identified *Timp1* and *Cd44* as the hub genes, which emerged as the most strongly up-regulated gene and most connected gene, respectively, at all post-lesion times. *Timp1* is an inhibitory molecule which regulates matrix metalloproteinases (MMPs), a disintegrin and metalloproteases (ADAMs) and ADAMs with thrombospondin motifs (ADAMTSs) [46]. In the context of a traumatic lesion of the CNS, MMPs including Timp1 are critical for synaptic recovery following axonal injury, mediating secondary degeneration and regulating angiogenesis and glial scar formation [47]. Timp1 also has pro-oligodendroglial properties in different in vitro contexts which promote OPC maturation [48] and in vivo inhibition of Timp1 activity completely abolishes spontaneous remyelination [49]. This protein is also associated with various integrins, including ItgaV, Itgb1 and Itgb3, which are involved in OPC differentiation into myelinating oligodendrocytes [50]. Our data shows that there is a strong increase in the translation of mRNA of *Timp1* especially in the segment rostral to the center of the lesion. Moreover, the western blot analysis of the same region of the spinal cord showed a marked decrease in Timp1 protein level in the cell, this could be coupled to the higher release of Timp1 in the extracellular environment, confirming an increased activation of this protein after SCI [51]. This protein, in fact, is a fundamental inducer for OPCs cell division and axonal regeneration after injury [52].

CD44, a non-kinase transmembrane glycoprotein, belongs to a family of cell surface glycoprotein receptors which are widely expressed in normal adult tissues, serving as adhesion molecules and mediating various biological processes, including wound repair and leucocyte trafficking [53]. CD44 is also required for migration of the myelin-repairing cells known as OPCs to focal inflammatory demyelinating lesions in the spinal cord [54]. Even if in the present study we focused only on the protein component of the ECM, Cd44 is a hyaluronic acid receptor and the hyaluronan component of the ECM is affected by the SCI, being also directly involved in the induction of astrocytes proliferation [55]. Thus, the regulation of *Cd44* expression by SCI suggests an involvement of the hyaluronan component remodeling.

Using a bioinformatic approach, we analyzed the differentially regulated genes in correlation with their role in the ECM protein-protein interaction net, finding a central role of *Timp1* and *Cd44*. The need of an informatic-based approach was also highlighted by lack of correlation between the single gene expression analysis and the motor-functional BBB score. The bioinformatic resources is able to expand the obtained results through algorithms involving protein-protein interactions from validated literature databases adding also different molecular players outside the analyzed genes net.

The bioinformatic prediction was then experimentally confirmed by the protein analysis of Timp1 in spinal cord tissue and by in vitro experiments using primary astrocytes, mainly involving Cd44 [56] and TIMP1- producing cells [57]; indeed when cultures of primary astrocytes are exposed to inflammatory stimuli, a strong activation in the expression level of both proteins is observed.

In conclusion, in this study, we demonstrated that a complex regulation of ECM composition, one which is both anatomically and temporally specific, takes place in the spinal cord segments outside the lesioned area following injury. This indicates that not only the ECM related to the scar formation but also the ECM adjacent to the lesion areas (interstitial, perineuronal, basement membrane), are profoundly altered in SCI. The temporal and anatomical profile of such regulation mechanisms should also be considered for regenerative medicine purposes, including the use of “smart” biomaterials.

## 4. Materials and Methods

### 4.1. Animals and Surgery

All animal protocols described herein were carried out according to European Community Council Directive 2010/63/EU and Italian legislation (Legislative Decrete 26/2014) and in compliance with the ARRIVE (Animal Research Reporting of In Vivo Experiments) guidelines and NIH Guide for the Care and Use of Laboratory Animals. The project has been reviewed by the Animal Welfare Body of IRET Foundation and approved by the Italian Ministry of Health (authorization no. 574/2015-PR of 22/06/2015).

Female CD-Sprague Dawley rats (200–250 g) were used in this study (Charles River Laboratories, Lecco, Italy). All animals were housed in pairs and had food and water ad libitum. On the day of surgery, the rats were pre-medicated with enrofloxacin and tramadol (5 mg/kg, s.c.), then anesthetized with isoflurane (1–3%) in O_2_, before undergoing a contusive spinal cord lesion at thoracic level T9. Briefly, the rats were immobilized on the stereotaxic table and a 4 cm longitudinal, median dorsal incision made from T8 to T10. The soft tissues were dissected layer by layer to fully expose the processus spinosus of T8 to T11 and the T9 processus spinosus and lamina were removed by clamp to expose the spinal canal and spinal dura. Spinal cord injury was performed using Impact One Impactor (Leica Biosystems, Wetzlar, Germany) at T9 level using a 1.5 mm tip with a force of 1 N (0.75 m/s) and 0 s of stance time and the depth of impact was 1.5 mm in order to reach the ventral horns of the gray matter. The back muscles were sutured and the skin incision closed with wound clips. The rats were treated with antibiotics and painkillers (enrofloxacin 5 mg/kg and tramadol 5 mg/kg s.c.) for 3 days post lesion and unlesioned animals were used as a control group.

The wellness of the rats was assessed by body weight monitoring and clinical score recording, evaluated daily for the first two weeks, then once a week until the day of sacrifice. Hind limb functional locomotor loss was evaluated using the BBB score [58] prior to the lesion, three days post-lesion, then once a week after surgery. Lesioned animals receiving a score greater than 10 at 3 DPL were excluded from the study.

On the day of sacrifice at 1, 8 or 45 DPL, the rats were euthanized with chloral hydrate (37%). The tissues of interest were dissected, immediately snap frozen and stored at −80 °C until use (gene expression study) or perfused with 4% paraformaldehyde and 14% picric acid in 0.2 M Sorensen buffer (pH 6.9) (histology study). The number of animals included in each experiment is indicated in the results section and in the legend to the figures.

### 4.2. Molecular Biology Analysis

For mRNA pathway array analysis, spinal cord segments of 1 cm length rostral and caudal to the lesion were collected, snap frozen and stored at −80 °C. The total RNA was extracted from the homogenized tissues using RNeasy Plus Universal Mini Kit (Qiagen, Hilden, Germany) according to manufacturer’s instructions. For cDNA synthesis, 5 µg of pooled RNAs were used from each experimental group. The cDNA was synthesized using the RT2 first strand kit (Qiagen). Eighty-four genes involved in the extracellular matrix were analyzed using the RT2 Profiler Extracellular Matrix and Adhesion Molecules PCR Array (Qiagen, PARN-013ZD). According to the manufacturer’s protocol, real-time PCR was performed using the RT2 SYBR Green qPCR Mastermix (Qiagen) with the CFX96 Touch Real Time PCR Detection System (Biorad, CA, USA).

For RT-qPCR analysis, cDNA synthesis of 1 µg of single sample RNA was performed using the iScriptg DNA Clear cDNA Synthesis Kit (Biorad), according to the manufacturer’s protocol. A total of 10 ng per sample of cDNA was used for each RT-qPCR for single gene analysis (see Table 1 for primer list). Amplification was performed using the SsoAdvanced Universal SYBR Green Supermix (Biorad) and CFX96 Touch Real Time PCR Detection System (Biorad). Relative quantification of mRNA was calculated using the comparative cycle threshold method. Ct values were collected for each gene analyzed, standardized on the *Rplp1* house-keeping gene and normalized on the respective intact segment of spinal cord. Gene expression fold change was then calculated as 2^(−ΔΔCt).

### 4.3. Bioinformatic Data Analysis

The data obtained from the PCR array was analyzed using the GeneGlobe platform (Qiagen) All PCR arrays were normalized on the same house-keeping gene (*Rplp1*) as suggested by the software and cut-off for Ct was set at 35. Gene expression variation was expressed as fold change with the ΔCt method, with respect to each segment from the intact animal group. To visualize the distribution of gene expression across the group and their clusterization, gene expression was plotted on a clustergram using the Cts values for each gene in the array and relative gene expression shown using heat maps of the PCR array. Genes with a fold change of ±2 or more were used for bioinformatic analysis. Panther software was used for clustering genes according to the biological function [59].

The Search Tool for the Retrieval of Interacting Genes (STRING v11) online database (http://string-db.org/) gives a critical assessment as well as the integration of PPIs, including experimental and predicted associations [60]. The Qiagen RT2 PCR array gene list was elaborated by the STRING software to outline protein-protein interactions (PPIs) of the ECM. Cytoscape v3.7.2 [61] is a widely used opensource software tool for the visualization of complex networks, with the ability to integrate all types of attribute data. Cytoscape was used to identify the hub genes, using a gene expression profile superimposed over the ECM network to eliminate nodes with fold regulation > ±2. This generated a specific network for each sample and the networks were continuously filtered based on degree parameter to give a maximum of 10 nodes. These genes were then used for further investigations.

### 4.4. Histology

After perfusion, the spinal cord tissue was dissected, post-fixed for 24 h, then washed in 5% sucrose in 0.2 M Sorensen buffer (pH 7.4). The spinal cords were collected and divided into 1 cm segments rostral and caudal to the lesion. Coronal (*n =* 3) and longitudinal (*n* = 3) cryostat sections (CM1950, Leica Biosystems) with a thickness of 14 µm were collected and processed for hematoxylin/eosin (H/E) and Nissl staining, to characterize the lesion extension at all time points. To define the lesion area, sections were captured using a Nikon Microphot—FXA equipped with a Nikon DXM1200F CCD camera (Nikon) at 4× magnification and reconstructed using Photoshop’s photomerge function (Adobe). The lesion area was then determined for each reconstructed section using ImageJ software (NIH) by calculating the number of pixels occupying the lesion site and different levels of the same spinal cord were aligned to give a 3D reconstruction (sampling step 210 μm).

### 4.5. Western Blot

Western blot analysis was performed to quantify Timp1 protein expression in the spinal cord at 1, 7 and 60 DPL. The rostral segments of spinal cord were homogenized according to RNeasy Mini Kit (Qiagen) protocol (with a ratio of mg of tissue to ml of lysis buffer, provided by the kit, equivalent to 1:10 weight/volume). Protein isolation was performed from the first flow-through of the spin column, by precipitation in acetone. Briefly, four volumes of ice-cold acetone were added to the flow-through from the RNeasy spin column, then incubated for 30 min at −20 °C. After spinning at 15,300× *g* for 10 min and removing the supernatant, the pellet was allowed to dry, then resuspended in RIPA buffer and protease inhibitor (1× Cocktail Sigma, 1 nM of PMSF, 10 nM of sodium fluoride, 1 nM of sodium orthovanadate).

Total protein concentration was estimated using a standard colorimetric method based on the Lowry assay (DC Protein Assay, Bio-Rad). For each sample, 40 µg of proteins and the marker protein (Precision Plus Protein Standards, Bio-Rad), diluted at a ratio of 1:100, was added to a solution of Laemmle/β-mercaptoethanol and after heating treatment (100 °C, 5 min), the proteins were resolved in 4–20% Mini-PROTEAN TGX Stain-Free Gels (Bio-Rad). Nitrocellulose Membrane (Bio-Rad AmershamProtran 0.45µm) has been used for protein transfer. A solution of 2.5% BSA in TBST (Tris Buffer Saline solution containing 1% Tween20) has been used for blocking. Incubation with the primary antibody (rabbit Timp1 Abcam ab61224, 1:500; mouse β-actin, Santa Cruz Biotechnology -Dallas, Texas, 1:200) has been performed overnight at 4 °C, whereas the incubation with HRP-conjugated secondary antibodies (swine anti-rabbit, Dako, 1:5000; swine anti-mouse, Dako, 1:5000) and HRP-conjugated protein for marker visualization (Precision Protein StrepTactin HRP-conjugate, Bio-Rad, 1:20,000) for 1 h at RT. Three washes with TBST was performed after incubation with antibodies, either the primary both the secondary. Clarity Western ECL Substrate (Bio-Rad −5 min incubation at RT in darkness) and the BioRadChemi DOC MP imaging system was used to detect immunoreactive signal.

The Fiji (ImageJ v2.1.0) software was used to measure densitometry. TIMP1 signal was normalized first on β-actin and then the ratio of lesioned (SCI+) to intact (SCI-) was calculated with samples present in the same gel.

### 4.6. Cell Cultures, Gene Expression Analysis and Immunocytochemistry

Primary astrocytes were isolated from 7-day-old mice using a standard protocol [62] and cultured in Dulbecco’s Modified Eagle’s Medium (DMEM) containing 15% fetal bovine serum (FBS), non-essential amino acid mixture (Sigma-Aldrich, St Louis, MO, USA) pen/strep (Invitrogen, Carlsbad, CA, USA) and 2 mM Glutamine (Invitrogen). Cultures were seeded in T25 flasks (125,000 cells/cm^2^) and maintained at 37 °C 5% CO2, detached with trypsin (10 min, 37 °C) and split twice before use.

When astrocytes shown 70% confluence the cytokine mix (TGF-β1, TNF-α, IL-1β, IL-6, IL-17 and IFN-γ; 20 ng/mL each; (Thermo Fisher Scientific, Waltham, MA, USA) or vehicle (0.04% of the cytokine solvent: 10% glycerol/100 nM glycine/25 nM Tris, pH 7.3) was added in the culture medium for 48 h. Cells were than lysed and the total RNA extracted using the RNeasy micro kit (Qiagen) following the manufacturer’s instructions and total RNA quantified by Nanodrop 2000 spectrophotometer. First strand cDNA was obtained using iScriptTM cDNA Synthesis Kit (Bio-Rad) following the manufacturer’s instructions and using a No-RT sample to check for genomic DNA contamination.

Semi-quantitative real-time PCR was performed using the CFX96 real-time PCR system (Bio-Rad) with 20 µL total volume, containing 1× SYBR Green qPCR master mix (Bio-Rad), 0.4 µM forward and reverse primers and 10 ng cDNA. The PCR reaction thermal profile consisted of 40 cycles of amplification (95 °C for 15 s and 60 °C for 60 s) after an initial denaturation step (95 °C, 2 min). Melting curves of the amplified products were obtained according to the following temperature/time scheme. The 2^(−ΔΔCt) method was used for the calculation of the relative gene expression, using *Gapdh* as house-keeping gene and the two isoforms of *Timp1* (a and b) and *Cd44* as investigated genes.

To check the quality of the cultures, a number of astrocytes were seeded on coverslips, fixed and stained for GFAP expression. In particular, the cells were fixed with ice-cold 4% paraformaldehyde for 20 min at RT. After two washes in PBS, the cells were incubated for 1 h with 1% BSA/1% donkey serum in PBS-0.3% Triton-X 100, then incubated overnight with primary antibody anti-GFAP (rabbit, Dako, 1:1000) diluted in PBS-0.3% Triton-X 100. The cells were then washed with PBS and incubated with secondary antibody goat Alexa 568-conjugated anti-rabbit (Invitrogen) for 30 min at 37 °C. After immunofluorescence staining, cells were incubated with the nuclear dye Hoechst 33,258 (1 μg/mL in PBS, 0.3% Triton-X 100) for 20 min at RT. As a final step, the cells were washed in PBS and coverslipped in glycerol and PBS (3:1, *v*/*v*) as mounting medium containing 0.1% paraphenylenediamine. A Nikon Eclipse E600 microscope (Nikon Instruments Europe BV, Amsterdam, Netherland) equipped with a QImaging Retiga 20002V digital CCD camera (QImaging, Surrey, BC, Canada) was used for the acquisition of representative images. 

### 4.7. Statistical Analysis

All statistical analyses were performed using GraphPad Prism v8.0 (GraphPad Software, San Diego, CA, USA). BBB scoring and animal weight were represented as mean ± SEM. For PCR array analysis, the data was reported as relative fold change expression. For the mRNA expression analysis obtained with qPCR, the data was reported as the Log2 of fold change and *p* values were calculated using one-way ANOVA on the ΔΔCts from each single animal, normalized on the mean of the ΔCt of the respective group of intact spinal cord segment_._ Uncorrected Fisher’s LSD post hoc analysis was used to compare the relative expression of each time point in the spinal cord segments rostral and caudal to the lesion and Dunnett’s post hoc analysis was used to compare the relative expression of lesioned animals compared to intact. *p* values inferior to 0.05 were considered significant.

## Figures and Tables

**Figure 1 ijms-22-01744-f001:**
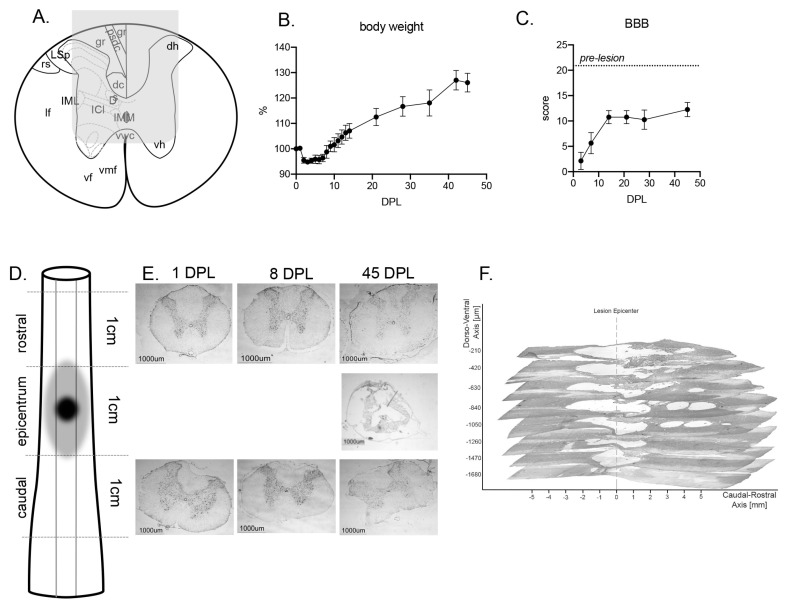
Functional and anatomical characterization of the traumatic spinal cord injury (SCI): (**A**) Gray area shows the tip of the impactor on the anatomical schema of the spinal cord; (**B**) Percentage variation of the body weight of lesioned rats over the course of the experiment. Data is expressed as mean ± SEM; (**C**) Basso, Beattie, Bresnahan (BBB) score in lesioned animals over the course of the experiment, where 21 is the score of healthy pre-lesion animals. Data is expressed as mean ± SEM; (**D**) Schematic diagram of spinal cord sampling; (**E**) Low-mag micrographs of the coronal section of the spinal cord at the center of the lesion, rostral and caudal levels (Nissl staining) from representative animals sacrificed at 24 h, 8 and 45 days; (**F**) 3D reconstruction of the spinal cord cavitation at 45 days post lesion (DPL) as derived from low-magnification micrographs of longitudinal sections, showing the rostro-caudal and dorso-ventral extension. Abbreviations: A; dc, dorsal corticospinal tract; dh, dorsal horn; Gr gracile fasciculus; ICI, Intercalated nucleus; IML, intermediolateral column; IMM, intermediomedial column; lf, lateral funiculus; LSp, lateral spinal nucleus; psdc, post synaptic dorsal column; vf, ventral funiculus; vh, ventral horn; vmf, ventral medial fissure; vwc, ventral white commissure.

**Figure 2 ijms-22-01744-f002:**
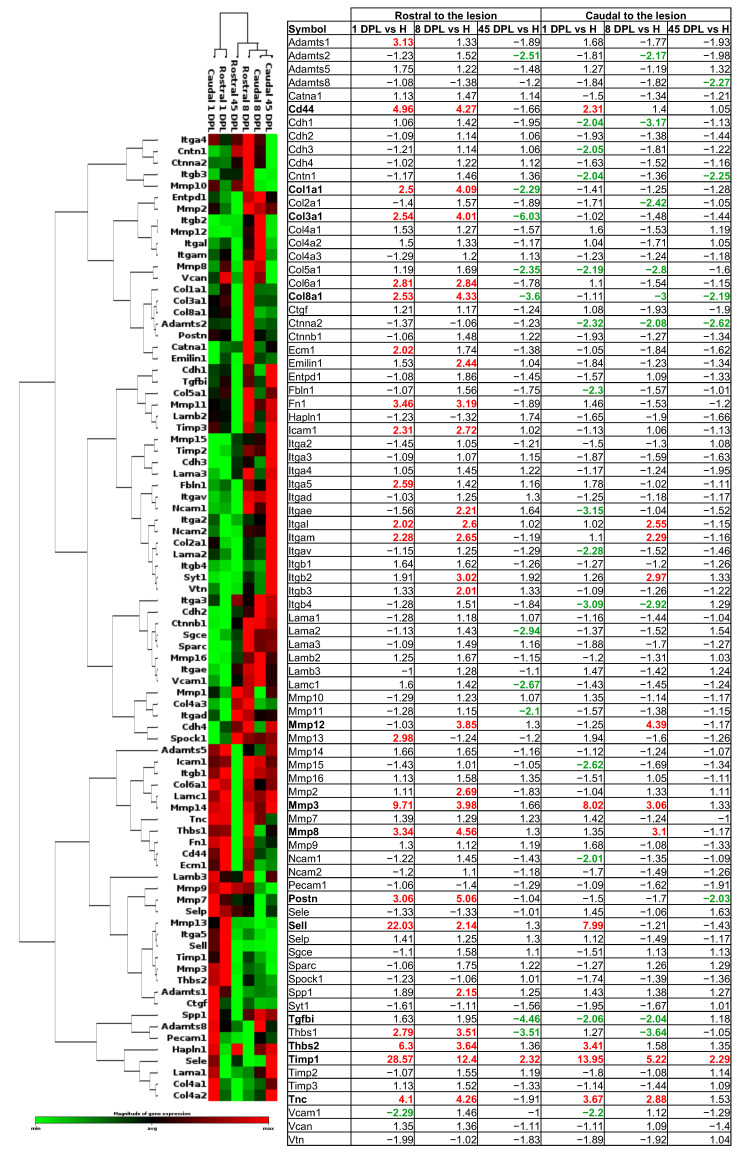
Heat map of the cluster analysis of extracellular matrix (ECM) gene expression regulation. ECM gene mRNA expression in the rostral and caudal segments of the spinal cord at all time points are shown in the table as fold change compared to the pool of intact segments of spinal cord (*n =* 5 per group, pooled). The clustergram shows the co-regulated genes across the group, analyzed with a vertical dendrogram and the clusterization of the segments of spinal cord at each time point illustrated with a horizontal dendrogram. A red color indicates a higher expression with respect to the median of the gene, whereas a green color indicates a lower expression. Gene names in bold indicate the highly regulated genes (fold of change > 4).

**Figure 3 ijms-22-01744-f003:**
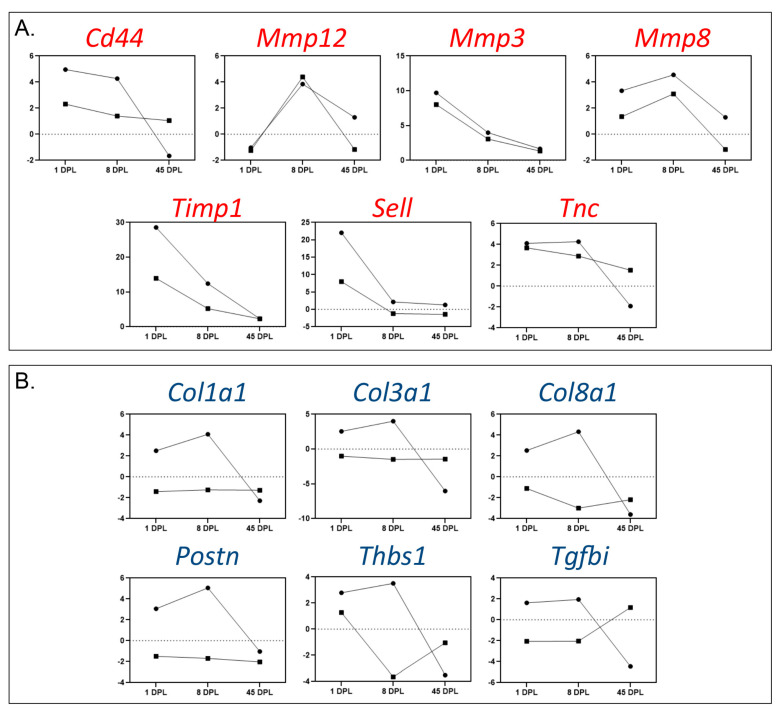
Time-course changes in the mRNA expression level of highly regulated genes. The figure illustrates the data obtained from the Polymerase Chain Reaction (PCR) array of genes regulated more than 4 times in at least one of the two segments and at least one considered time point. Genes are divided into similarly (**A**) or differently (**B**) regulated in the rostral and caudal segments. Circles represent the segment rostral while squares the region caudal to the lesion.

**Figure 4 ijms-22-01744-f004:**
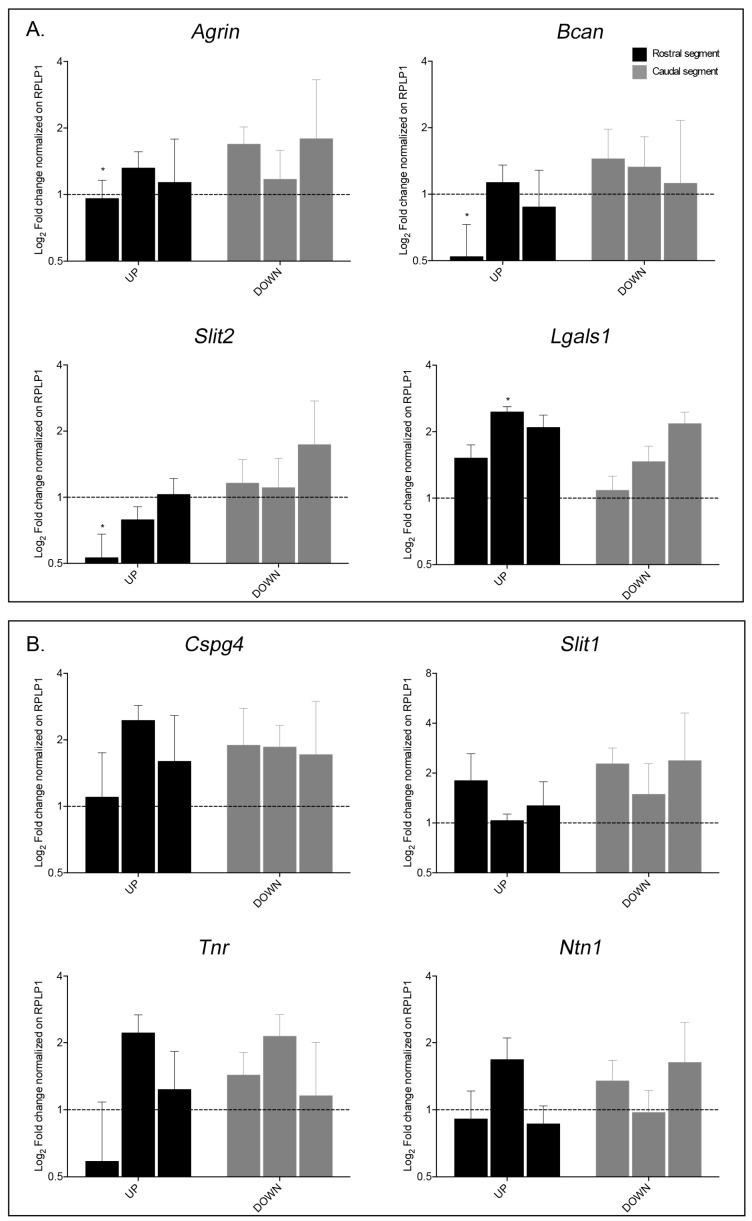
Time-course changes in mRNA expression level for neural extracellular matrix encoding genes: (**A**) Genes presenting a statistically significant difference in expression level; (**B**) Genes presenting non-significant variation. Real Time – Polymerase Chain Reaction (RT-PCR) data is represented as Log2 fold change of gene expression normalized on the respective intact segment of spinal cord, shown by the dotted line (*n =* 4 animals per group per time point). The black bars show the results obtained from the rostral segment, whereas the gray bars show the segment caudal to the lesion. Statistical analysis: one-way ANOVA with Fisher’s LSD, *Agrin p =* 0.041, t = 2.146 (4, 27); *Bcan p =* 0.0441, t = 2.112 (5, 27); *Slit2 p =* 0.0341, t = 2.232 (5, 27); *Lgals1 p =* 0.035, t = 2.22 (4, 27), * *p <* 0.05.

**Figure 5 ijms-22-01744-f005:**
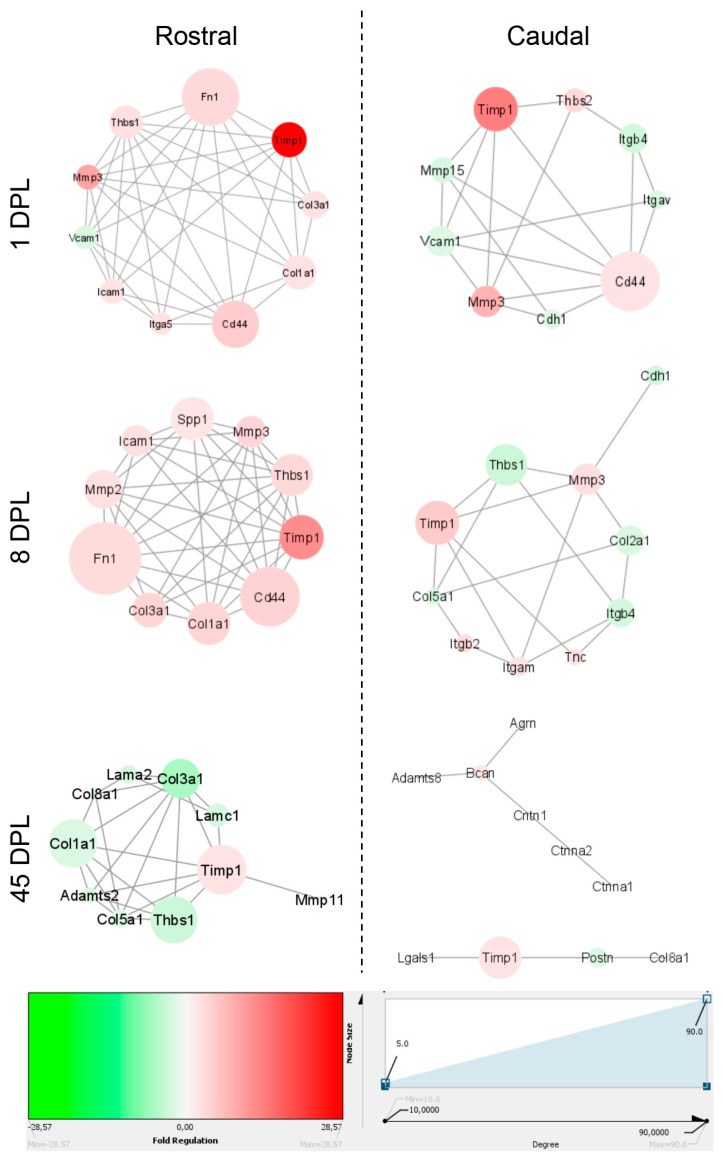
Cluster analysis of the investigated genes. Interaction network resulting from degree and fold change analysis of the general Extracellular Matrix (ECM) network. Colors indicate gene expression as fold regulation, where red indicates up-regulation and green down-regulation. The size of the nodes represents the degree of gene involvement in the ECM network. The edges represent the interactions between each node.

**Figure 6 ijms-22-01744-f006:**
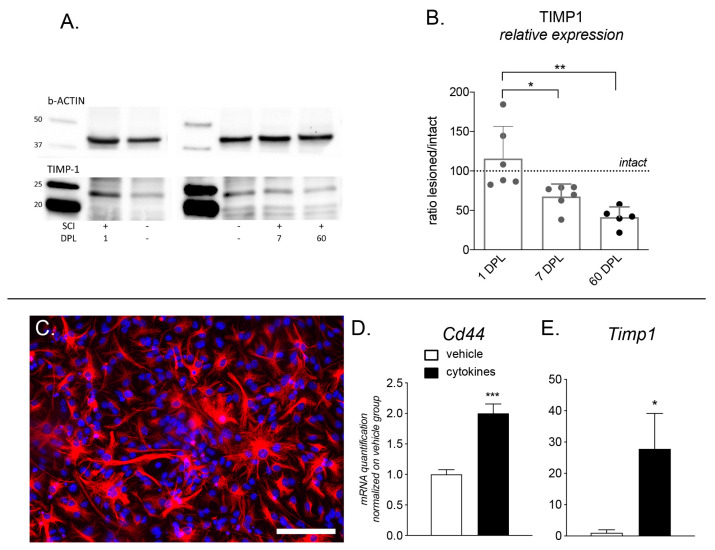
Timp1 protein quantification in the spinal cord and expression of *Cd44* and *Timp1* in primary astrocyte cultures: (**A**) Two representative gels are shown in the figure. Upper panel shows b-Actin (43 kDa), used as house-keeping protein, lower panel shows Timp1 (23 kDa) in each analyzed group (intact animals: SCI-; lesioned: SCI+ at 1 DPL, 7 DPL and 60 DPL); (**B**) Graph shows relative *Timp1* expression calculated as signals as measured by densitometry, indicated by relative intensity (lesioned versus controls). Statistical analysis: one-way ANOVA and Tukey post-hoc test, * *p <* 0.05; ** *p <* 0.001; (**C**) Micrograph illustrates pure astrocyte cultures stained for GFAP marker; (**D**,**E**) Graphs show mRNA expression level of *Cd44* (**D**) and *Timp1* isoform b (**E**) in cultures exposed to vehicle or cytokine mix for 48 h. Statistical analysis. Student’s *t*-test. Asterisk represents difference between vehicle- and cytokine- treated groups (* *p* > 0.05).

**Table 1 ijms-22-01744-t001:** Sequences and genetic loci of primers used for RT-qPCR analysis.

Gene	Specificity	Genetic Locus	Forward Sequence (5’–3’)	Reverse Sequence (5’–3’)
*Acan*	rat	NM_022190.1	GTGAGATCGACCAGGAGCCA	TCGGGAAAGTGGCGATAACA
*Agrin*	rat	NM_175754.1	CCTGCAACATCTGCTTGATCC	GGATTCCAGGTTTGTAGTTGCTG
*Bcan*	rat	NM_001033665.1	GGACCTCACAAGTTCTTCCAAGT	CTTTCAGGTCATCAGCGAGGG
*Cd44*	rat	NM_012924.2	AACTACAGCCTTGATGACTACCC	ATGACTCTTGGACTCTGATGGTT
*Cd44*	mouse	NM_009851.2	AGAAGAGCACCCCAGAAAGC	CTTGCAATGGTGGCCAAGG
*Cspg4*	rat	NM_031022.1	AACAGGAAAAAGCACCCCCA	ACCTGTCTTGTTGCGTTTGC
*Fn1*	rat	NM_019143.2	AAGACAGATGAGCTTCCCCAA	TGAACTGTGGAGGGAACATCC
*Gapdh*	rat/mouse	NM_001113417.1	GGCAAGTTCAATGGCACAGTCAAG	CATACTCAGCACCAGCATCAC
*Lgals1*	rat	NM_019904.1	TTCAATCATGGCCTGTGGTCT	CTCTCCCCGAACTTTGAGACA
*Ntn1*	rat	NM_053731.2	AGGACTATGCTGTCCAGATCCA	TACGACTTGTGCCCTGCTTG
*Postn*	rat	NM_001108550.1	TGCAAAAAGACACACCTGCAAA	GGCCTTCTCTTGATCGCCTT
*Rplp1*	rat	NM_001007604.2	GGCAGTCTACAGCATGGCTT	GTTGACATTGGCCAGAGCCT
*Sell*	rat	NM_019177.3	ATCGCAGGAAAGGATGGATGAT	GGTTTTTGGTGGCGGTTGTT
*Slit1*	rat	NM_022953.2	CGCAAGGGCGCATCGT	GGGGCTATCTCCAGGTGCTAT
*Slit2*	rat	NM_022632.2	GGGGCCATAATGTAGCAGAGG	GACTGGTGACCTTCTTCCTCA
*Tnc*	rat	NM_053861.1	ATTGTCTACCTCTCTGGAATTGCTC	TTCCGGTTCAGCTTCTGTGG
*Tnr*	rat	NM_013045.1	CCTCAATGGGGAGTTAAGCCA	CTGGAAAACAATCCAGCCGC
*Timp1-a*	mouse	NM_011593.2	TGGGTGGATGAGTAATGCGTC	GGCCATCATGGTATCTCTGGT
*Timp1-b*	mouse	NM_001294280.2	CAACTCGGACCTGGATGCTAA	ACTCTTCACTGCGGTTCTGG

## Data Availability

The data presented in this study are available on request from the corresponding author.

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
