# Peer review of "Time-Course Changes of Extracellular Matrix Encoding Genes Expression Level in the Spinal Cord Following Contusion Injury—A Data-Driven Approach"

_ijms, 2021, doi:10.3390/ijms22041744_

Round 1

Reviewer 1 Report

This is a potentially interesting work that explores the expression profile of extracellular matrix-related genes in a rat model of spinal cord injury (SCI). Since the degeneration process that ensues after SCI is asymmetrical (rostral respect to caudal areas), the authors explore this anatomical difference in their analysis as well. The authors then confirm some of the in vivo gene expression results by analysing protein levels in vivo and gene expression in vitro after inflammatory challenge. The in-vitro experiment is misleading and does not contribute to the overall conclusions of the manuscript. On the contrary, the novelty of the paper comes from the meta-analysis the authors perform on the gene expression results, finding relationships and connections between the different up and downregulated genes. This meta-analysis, although powerful, is under-discussed. The text is clear and correct, though the discussion could be improved and the figures need reworking regarding resolution and readability. I have the following comments:

Major

1. The species (rat) should be mentioned in the abstract.

2. Function of Timp1 and Cd44 should be stated in the abstract. (e.g. “hyaluronan-receptor CD44 and MMP-inhibitor Timp1…”)

3. In the abstract, perhaps it is more important to mention what do the up/downregulated genes do (ECM remodelling, ECM adhesion), than how many genes have changed, to highlight the importance of the findings. Also mentioning the asymmetry in ECM gene expression is important in the abstract.

4. Why only female rats were used?

5. Lines 56 to 58: MMPs degrade a variety of proteins, but do not degrade hyaluronan (a sugar polymer). Hyaluronan is degraded mostly by hyaluronidases either in the cell surface or inside lysosomes, and also by reactive oxygen species when injury (Garantziotis and Savani, 2019, Matrix Biology). As Gaudet & Popovich comment in their 2014 review on SCI and ECM (cited by the authors in this sentence), the suggested mechanism would be the opposite: Hyaluronan fragmentation during injury generates small HA fragments (LMW-HA), which may act as a DAMP and stimulate MMP activity and/or MMP expression.

6. Among the additional genes tested by RT-qPCR, genes related to hyaluronan metabolism (Has, and hyals) should be tested as well. Changes in the hyaluronan scaffold do not always correlate with changes in the protein components of the neural matrix (CSPGs, HSPGs, etc). Since CD44 levels are altered, changes in hyaluronan are also expected. The paper in its current form focuses only in the protein components of the extracellular matrix, disregarding completely the glycan scaffold (hyaluronan). If this experiment is not performed, this limitation should be mentioned in the discussion.

7. The absence of MMP1 expression after SCI is contradictory with other reports in rat (Zhou 2014 Cell Mol Neurobiol) and human post-mortem samples (Buss 2007 BMC Neurol). This should be discussed.

8. The in vitro analysis in Fig 6, although correctly done, does not necessarily mean that the increase in CD44 and TIMP observed in vivo after lesion is carried out solely by astrocytes. In fact, in inflammatory context, CD44 is also highly upregulated both in microglia and infiltrating macrophages (Mrdjen 2018 Immunity). Double immunolabelling or gene expression analysis of FACS-sorted cells (or expensive single-cell sequencing) could better answer this question. The in vitro analysis, thus, is insufficient to claim that astrocytes are responsible for the changes observed in vivo, since the authors use a pure astrocyte culture, disregarding the rest of CNS cells. This whole panel could be placed as supplementary, since among the main figures it might be misleading.

9. It is not clear what additional information is obtained from the meta-analysis of gene expression (Fig 5). Both the results and discussion that mention this figure do so in relation to up or down regulation only, information already reported in figs 3 and 4.

Figures

-Figures 1E and 1F are of low quality/low resolution. In Fig 1E, the lettering is blurred, and in Fig 1F the axes are difficult to read due to low resolution. It could be due to compression from the pdf format, but nonetheless it must be addressed.

-Fig 3: It should be stated in the figure legend which symbol corresponds to rostral and which to caudal.

-Fig 4: Font of x and y axis too small. Please increase font size.

-Fig 5: Again problem of resolution (or compression). Pixelated/blurry images complicate reading the gene names.

Minor

-line 43, typo: “to a fibrotic”

-line 354, typo: “Timp1 are”

-Both CD44 and tenascins levels remain high at 8 DPL. Since CD44 is reported to be implicated in migration (among other things) and higher levels of tenascins correlate with more stable 3D scaffold structure (Richter 2018), perhaps this helps migration alongside CD44?

-TIMP1 levels are 20 fold higher after injury. This might be implicated in cell proliferation (inflammation)?

-I understand that pooling was done to reduce variance, but it would have been interesting to assess the expression profile per individual and correlate with the degree of the lesion (the BBB score).

-The discussion is written in a way where it is not clear which results are from this paper and which are from previous studies (e.g. lines 318 to 322, where “other proteins not related to the ECM” seem to have been evaluated in this paper, whereas it has been done in refs 26-28).

Reviewer 2 Report

This is a useful study providing data on ECM gene regulation after spinal cord injury
with respect to location in the injured cord. This is a novel aspect. Data is generated
by gene array and confirmed by qRT-PCR and Western blot (for Timp1). The authors
also show that astrocytes in vitro can be induced to express ECM molecules with
cytokines, partly mimicking the injury situation.
Here are a few points for the authors to address:
1) In Fig.2, it is not clear what the regulation is compared to – unlesioned spinal
cord? In general, the authors should expand on the rationale for comparisons and
time points chosen in the results.
2) The authors use the term ‘epicentrum’ to describe the site of injury. Should this
not be called “centre of the injury”?
3) “Sections collected at the epicentrum at 1 and 8 DPL were almost destroyed…”
So what was the cellular composition at that time point? Could the apparent massive
upregulation at 1 dpl of some genes simply be a consequence of sampling other
tissues, e.g. meninges or connective tissue?
4) “We first confirmed the data obtained using PCR arrays for CD44, Fn1, Postn, Sell
and Tnc, showing a strong up-regulation in both segments at 1 and 8 DPL (data not
shown).” This data should be shown as independent confirmation of array data.
5) The authors do not observe regulation of cspg4. Is that at odds with the literature?
6) The bioinformatics in Fig. 5 is unclear. The authors state that “The size of the
nodes represents the degree of gene involvement in the analyzed cellular process.”
What cellular process is meant? The discussion should be more explicit about what
this analysis contributes to the paper.
7) Fig. 6D – show individual data points.

Author Response

Reviewer 2

This is a useful study providing data on ECM gene regulation after spinal cord injury with respect to location in the injured cord. This is a novel aspect. Data is generated by gene array and confirmed by qRT-PCR and Western blot (for Timp1). The authors also show that astrocytes in vitro can be induced to express ECM molecules with cytokines, partly mimicking the injury situation.

Here are a few points for the authors to address:

1) In Fig.2, it is not clear what the regulation is compared to – unlesioned spinal cord? In general, the authors should expand on the rationale for comparisons and time points chosen in the results.

We apologize if the figure caption is not clear to the Reviewer, however it is stated “ECM gene mRNA expression in the rostral and caudal segments of the spinal cord at all time points are shown in the table as fold change compared to the pool of intact segments of spinal cord”.

The three time points included in the study correspond to the inflammatory phase (24 hours post-lesion), ongoing secondary degeneration (8 days post-lesion), and the chronic phase (45 days post-lesion), as stated in the introduction (line 94 – 97).

In the revised version we verified that all the comparisons are clearly stated for each analysis both in results section and figure legends.

2) The authors use the term ‘epicentrum’ to describe the site of injury. Should this not be called “centre of the injury”?

We agree to change the terminology.

3) “Sections collected at the epicentrum at 1 and 8 DPL were almost destroyed…” So what was the cellular composition at that time point? Could the apparent massive upregulation at 1 dpl of some genes simply be a consequence of sampling other tissues, e.g. meninges or connective tissue?

We focused the molecular analysis on the spinal cord segments adjacent to the center of the injury, where the histological analysis indicates that tissue is present and conserved. This is detailed in the text (see Fig. 1 for tissue sampling),

4) “We first confirmed the data obtained using PCR arrays for CD44, Fn1, Postn, Sell and Tnc, showing a strong up-regulation in both segments at 1 and 8 DPL (data not shown).” This data should be shown as independent confirmation of array data.

Data have been added to supplementary materials (new Figure S1).

5) The authors do not observe regulation of cspg4. Is that at odds with the literature?

For the data presented in Figure 4 showed only the comparison between rostral and caudal segment, by using a One Way ANOVA with Fisher’s LSD (multiple t-tests between the rostral and caudal segments at the same time point).

We did no shown the differences between rostral OR caudal segments compared to the respective intact segments (dotted line).  

In line with the literature, we did observe a regulation of Cspg4 after spinal cord injury in the rostral segment compared to the control group at 8 DPL (Figure 4; Cspg4 graph, second black column t-test P = 0.0098).

6) The bioinformatics in Fig. 5 is unclear. The authors state that “The size of the nodes represents the degree of gene involvement in the analyzed cellular process.” What cellular process is meant? The discussion should be more explicit about what this analysis contributes to the paper.

We apologize if the figure is not well explained. The “cellular process” is the category of the algorithm used to build the net of the protein-protein interaction. As explained in the methods (line 621 - 628), in this case the genes in the arrays were used as input in the STRING software to generate the net uploaded in cytoscape software to measure the degree of the nodes. We added now the information in the figure legend and we added more information in the discussion (line 533 - 534).

Moreover, we added a correlation analysis between the single gene qPCR analysis and the BBB score, finding no significance (see supplementary table S1 and line 535 – 539). This negative result highlights the importance of an informatic-based approach, which allows the expansion of the analysis through protein-protein interaction nets based on validated databases, thus taking into account a more complex picture not limited to the genes of interest.

7) Fig. 6D – show individual data points.

As the 2^(ΔΔCt) method was used it is not possible to show the individual data of the relative expression. We added in the supplementary material the individual ΔCt values for each group and gene (Figure S3).

Reviewer 3 Report

The authors of the manuscript present an interesting study describing gene expression changes of extracellular matrix encoding genes in spinal cord affected by contusion injury in rats. Novelty of the work is based on the analysis of rostral and caudal segments adjacent to the lesion site. The authors described differential expression of many genes with respect to the control condition, but also with respect to the orientation to the site of spinal cord injury. The primary array-based analysis is further extended by characterization of few extra genes by individual RT-qPCR reactions, protein-protein-interaction analysis identifying hub genes, and western blot analysis of selected targets. The relevance of the most differentially expressed genes is commented in the discussion.

I appreciate the concept, introduction part, characterization of experimental model (first chapter and Fig. 1) and language that makes the manuscript easily readable. However, although I found the topic of general interest, the study is very poorly designed. The most problematic point is a lack of replicates in the array analysis that prohibits to evaluate statistical significance of obtained results. Without replicates, it cannot be evaluated the variability between animals, existence of outliers, their influence on the analysis etc. Although the authors used for the analysis a pool of five animals and justified it as a standard setting in array based experiment, I cannot accept this argument. Such setting may be justifiable for screening studies, where the primary analysis is used to prioritize targets that are further validated in follow-up experiments. This part is completely missing in the manuscript, on the contrary all analysis are based on this screening type of experiment, therefore may be substantially confounded by biological as well as technical variables that cannot be monitored in used experimental setting. From the technical point of view, I miss the description how the pool of five animals was prepared, how the quality of RNA was assessed, or how the stability of the reference gene that was used for normalization was evaluated. These raise additional doubts on the technical soundness of the presented data. Without any validation experiments showing the validity and reproducibility of the data in independent sample cohort, I cannot recommend the manuscript for publication.

Author Response

Reviewer 3

The authors of the manuscript present an interesting study describing gene expression changes of extracellular matrix encoding genes in spinal cord affected by contusion injury in rats. Novelty of the work is based on the analysis of rostral and caudal segments adjacent to the lesion site. The authors described differential expression of many genes with respect to the control condition, but also with respect to the orientation to the site of spinal cord injury. The primary array-based analysis is further extended by characterization of few extra genes by individual RT-qPCR reactions, protein-protein-interaction analysis identifying hub genes, and western blot analysis of selected targets. The relevance of the most differentially expressed genes is commented in the discussion.

1) I appreciate the concept, introduction part, characterization of experimental model (first chapter and Fig. 1) and language that makes the manuscript easily readable. However, although I found the topic of general interest, the study is very poorly designed. The most problematic point is a lack of replicates in the array analysis that prohibits to evaluate statistical significance of obtained results. Without replicates, it cannot be evaluated the variability between animals, existence of outliers, their influence on the analysis etc. Although the authors used for the analysis a pool of five animals and justified it as a standard setting in array based experiment, I cannot accept this argument. Such setting may be justifiable for screening studies, where the primary analysis is used to prioritize targets that are further validated in follow-up experiments.

We thank the Reviewer for his appreciation of the concept and the manuscript writing. However, the use of a pool as a screening method for arrays is totally in line with the whole concept of the study.

It is widely demonstrated that RNA pooling is a powerful method both for microarrays (Zhang et al., 2007) and sequencing (Assefa et al., 2020), optimizing costs and data generation.

In particular it is widely accepted that pooling increases the efficiency and the significant of the results without any adverse effect on the inference, concluding that “the realized benefits do not outweigh the price paid for loss of individual specific information” (Kendziorski et al., 2005).

The removal of individual variability, in fact, increases the possibility to identify the differentially regulated genes in the studied condition, especially for complex disease models as spinal cord injury. As it is possible to see from the main results, the identified genes are in line with the literature, adding new and specific information about the distance from the center of the lesion and using a bioinformatic approach, coupled with in vitro experiments, to confirm and expand data from the arrays.

Moreover, very stringent criteria are used for the inclusions of the animals in the pool and for the quality control of the PCR arrays reaction, with standard and validated procedures available from the manufacturer.

For the array analysis, the pooling is performed at the reverse transcription step, therefore individual extracted RNAs are used to validate arrays data (Figure S1) and perform qPCR experiments (Figure 4), again proving the robustness and the reliability of the data obtained from the pool.

We are aware that the Reviewer will not accept these arguments: this is an all-or-null discussion dividing scientist favorable to the pool strategy for transcriptome screening, and unfavorable. We then defer to Editor’s judgment.

Refs:

- Assefa et al. On the utility of RNA sample pooling to optimize cost and statistical power in RNA sequencing experiments. BMC Genomics, 2020, 21: 312.

- Kendziorski et al. On the utility of pooling biological samples in microarray experiments. PNAS, 2005, 102: 4252 – 4257.

- Zhang et al., Pooling mRNA in microarray experiments and its effect on power. Bioinformatics, 2007, 23: 1217.

2) This part is completely missing in the manuscript, on the contrary all analysis are based on this screening type of experiment, therefore may be substantially confounded by biological as well as technical variables that cannot be monitored in used experimental setting.

Data on the main identified genes were confirmed by qPCR experiments using single samples, as already stated in the text (line 196) and now these data have been added in the supplementary material (Figure S1).

3) From the technical point of view, I miss the description how the pool of five animals was prepared, how the quality of RNA was assessed, or how the stability of the reference gene that was used for normalization was evaluated. These raise additional doubts on the technical soundness of the presented data.

The description of the samples and the molecular biology analysis is already present in the methods section, line 444: “For mRNA pathway array analysis, spinal cord segments of 1 cm length rostral and caudal to the lesion were collected, snap frozen, and stored at -80°C. The total RNA was extracted from the homogenized tissues using RNeasy Plus Universal Mini Kit (Qiagen, Hilden, Germany) according to manufacturer’s instructions. For cDNA synthesis, 5 µg of pooled RNAs were used from each experimental group.”

And line 466: “The data obtained from the PCR array was analyzed using the GeneGlobe platform (Qiagen) All PCR arrays were normalized on the same house-keeping gene (Rplp1) as suggested by the software, and cut-off for Ct was set at 35.”

Quality controls for samples quality, housekeeping genes, inter- and intra-assay variability are included in the PCR arrays and related software, a widely used and validated technology.

4) Without any validation experiments showing the validity and reproducibility of the data in independent sample cohort, I cannot recommend the manuscript for publication.

Again, as explained for point 2, data were validated.

Round 2

Reviewer 1 Report

The authors have addressed most of the issues raised in the revision. The changes in the abstract and additions to the discussion are welcomed. Thanks also for the explanation about the usage of female rats, the infection problem makes sense.

The manuscript is now suitable for publication.

Notes:

-line 25. “a higher”

-I was not able to fully see the figures in the new pdf. I trust that the authors have improved image resolution and font size.

Author Response

REVIEWER 1

The authors have addressed most of the issues raised in the revision. The changes in the abstract and additions to the discussion are welcomed. Thanks also for the explanation about the usage of female rats, the infection problem makes sense.

The manuscript is now suitable for publication.

Notes:

-line 25. “a higher”

-I was not able to fully see the figures in the new pdf. I trust that the authors have improved image resolution and font size.

We thank the Reviewer for the useful suggestions which helped to improve the manuscript.

We corrected the typo.

We ensure that we uploaded high quality images in the manuscript, and we will forward the original figures in the final documents upload.

Reviewer 3 Report

The authors of the manuscript replied to majority of my comments. They provided justification of their experimental design, supported by relevant citations and extended the manuscript for an extra supplementary figure.

I appreciate references provided by authors that justify the pooling strategy for the array experiment that served, as confirmed by authors, as a screening. I do not have any further objection to this part. However, I still lack data and experimental details confirming the validity of the screening data. The supplemental figure 1 showing expression of four genes measured as in the array so as in the qPCR experiment does not seem to follow the identical trend (although authors claim so). Whereas in the qPCR experiment the four genes show upregulation almost in all time-points, downregulation can be often observed in the array experiment. To confirm if the data generated on different platform are in agreement or not (and are therefore valid), I encourage the authors correlate fold-changes and calculate significance of the correlation and provide such figure as a supplement as well. I would also appreciate to add qPCR validation (or another) of other genes which are discussed in the main text solely based on the array data. I also encourage authors to add details if the data was generated on identical or different sample set.

Regarding my original comments on the qPCR (assessment of RNA quality, selection of reference genes for normalization), I encourage authors to provide details on quality controls for samples, housekeeping genes, inter- and intra-assay variability that were included on PCR arrays as stated by authors. I also encourage the authors to check the MIQE Guidelines (Minimum Information for Publication of Quantitative Real-Time PCR Experiments) published in 2009.

Round 3

Reviewer 3 Report

The answers provided by authors did not confer my doubts about the validity of presented data.

Firstly, the main dataset is generated on sample pools consisted of five animals. As RNA quality of individual samples in the pool was not assessed, it raises questions if the overall signal could not be given by a single outlier. Although this risk may be excluded by relatively low SD values (?) measured on the same set of samples in qPCR measurements, number of assayed genes is relatively low to make such judgement for majority 84 genes measured by qPCR array. In addition, although the authors provided a new table comparing signals measured in the array and qPCR measurements to demonstrate the concordance of both measurements, they excluded some of the signals showing significant DE (i.e. logFC>+-2) in the array, but not in qPCR measurement (Postn, DOWN, 45 DPL) and vice versa (CD44, DOWN, 8 and 45 DPL etc.), i.e. those showing discordant trends. Authors also did not answer my repeated question on qPCR data normalization and selection of reference genes. Overall, this does not support the reliability of the data analysis process and reporting of its results.

Secondly, regardless of its reliability, the qPCR measurement is only a technical validation of the array experiment. It does not validate the biological findings, which is typically done for selected targets in the independent sample cohort. Such analysis would reject any doubts discussed in the previous paragraph and support the conclusion of the study. For now, I cannot see any warranty that if the authors take another (independent) five samples, they would get concordant result with the first analysis.

To sum up, due to my concerns on the validity of the array data and because authors did not provided any validation data, I cannot recommend the acceptance of the paper.

Author Response

The answers provided by authors did not confer my doubts about the validity of presented data.

Firstly, the main dataset is generated on sample pools consisted of five animals. As RNA quality of individual samples in the pool was not assessed,

As we explained several times and as it is also clearly written in the main text the RNA was extracted from single animals, so it was quantified and quality checked for each individual RNAs (Line 583: “The total RNA was extracted from the homogenized tissues using RNeasy Plus Universal Mini Kit (Qiagen, Hilden, Germany) according to manufacturer’s instructions. For cDNA synthesis, 5 µg of pooled RNAs were used from each experimental group.” Line 598: “For RT-qPCR analysis, cDNA synthesis of 1 µg of single sample RNA was performed using the iScriptg DNA Clear cDNA Synthesis Kit (Biorad), according to the manufacturer’s protocol”).

The pool was generated at the step of the reverse transcription and, again, the quality of the reaction was analyzed as already discussed in the previous answer, definitively proving the good quality of the samples, the reliability of the retrotrascription and the PCR reactions.

As specified in the MIQE guidelines cited by the Reviewer, we followed the main key points described in section 2:

  • Analytical sensitivity
  • Analytical specificity
  • Accuracy
  • Repeatability
  • Reproducibility

The firs 3 points are guaranteed by the PCRarray technique (Qiagen) which uses primers that are validated and standardized. Repeatability was validated by internal control in the PCR arrays and Reproducibility was validated by 6 different genes in qPCR (independent technique on the same samples).

As suggested by the MIQE guidelines in section 5.1, we performed the RNA quantification by Nanodrop (ThermoScientific) automatically checking also the quality of the extracted RNA (A260/A280 ratio). Genomic contamination, as already demonstrated in the revision 2, was checked during the PCR array by internal control, and also during qPCR by using a No-RT sample.

it raises questions if the overall signal could not be given by a single outlier.

As discussed in the previous revisions, this is exactly why scientists perform pool analysis: to decrease the variability and the effects of outliers. Again, the removal of individual variability, increases the possibility to identify the differentially regulated genes in the studied condition, especially for complex disease models as spinal cord injury. The identified genes are, in fact, in line with the literature.

Although this risk may be excluded by relatively low SD values (?) measured on the same set of samples in qPCR measurements, number of assayed genes is relatively low to make such judgement for majority 84 genes measured by qPCR array.

We believe that this comment reflects the negative idea of the Reviewer about the pooling that, as we discussed in the previous revisions, is a black-or-white view. In our opinion it is meaningless to perform all the validations for all the genes in the array. This will totally eliminate the meaning of the array.

We proved by qPCR in the same sample, using different cDNA synthesis from single animals, different primers and reagents the regulation of the two main genes that we discussed and throughout the whole text.

In addition, although the authors provided a new table comparing signals measured in the array and qPCR measurements to demonstrate the concordance of both measurements, they excluded some of the signals showing significant DE (i.e. logFC>+-2) in the array, but not in qPCR measurement (Postn, DOWN, 45 DPL) and vice versa (CD44, DOWN, 8 and 45 DPL etc.), i.e. those showing discordant trends.

As the Reviewer may have noted, there was no statistical significance in the columns, that in a scientific meaning there were no changes. Again, this is the reason why to use the pool strategy in this complex screening condition, as commented and published by many groups:

Again, in fact, pooling increases the efficiency and the significant of the results without any adverse effect on the inference, concluding that “the realized benefits do not outweigh the price paid for loss of individual specific information” (Kendziorski et al., 2005).

Refs:

- Assefa et al. On the utility of RNA sample pooling to optimize cost and statistical power in RNA sequencing experiments. BMC Genomics, 2020, 21: 312.

- Kendziorski et al. On the utility of pooling biological samples in microarray experiments. PNAS, 2005, 102: 4252 – 4257.

- Zhang et al., Pooling mRNA in microarray experiments and its effect on power. Bioinformatics, 2007, 23: 1217.

Authors also did not answer my repeated question on qPCR data normalization and selection of reference genes. Overall, this does not support the reliability of the data analysis process and reporting of its results.

Data normalization was automatically chosen by the GeneGlobe software as stated in the main text. “Line 611: The data obtained from the PCR array was analyzed using the GeneGlobe platform (Qiagen) All PCR arrays were normalized on the same house-keeping gene (Rplp1) as suggested by the software, and cut-off for Ct was set at 35.”

The same normalization was performed for the qPCR analysis. “Line 603: Ct values were collected for each gene analyzed, standardized on the Rplp1 housekeeping gene and normalized on the respective intact segment of spinal cord. Gene expression fold change was then calculated as 2^(-ΔΔCt).”

The software automatically chose the hosekeeping gene from the whole list of the analyzed genes, according to the variability between groups and the Ct. This is in line with the MIQE guidelines (section 8.1).

Secondly, regardless of its reliability, the qPCR measurement is only a technical validation of the array experiment. It does not validate the biological findings, which is typically done for selected targets in the independent sample cohort. Such analysis would reject any doubts discussed in the previous paragraph and support the conclusion of the study. For now, I cannot see any warranty that if the authors take another (independent) five samples, they would get concordant result with the first analysis.

We should remember to the Reviewer that the “samples” in the study are experimental animals, with a heavy experimental model (i.e. spinal cord injury). Building a new independent cohort with the same numerosity to validate data that are already robust, as accepted by the previously discussed literature, will be against all the animal experimentation rules and the 3Rs principles.

We would like to remember, also, that the pooling strategy is widely accepted also by the present journal, as reported in the following examples deriving from different kind of studies (in vitro, human samples)

> We want also to point out again that this is a common strategy also for in vitro experiments, as demonstrated by articles recently published in this journal (Łuczkowska et al., 2020). In fact, even starting from in vivo model and a more complex condition, we followed the same workflow: i) array screening; ii) validation of few selected genes; iii) bioinformatic analysis; iv) western blot. Moreover, we also added in vitro validation of the same identified gene targets.

Thus, our study does include technical and functional validation.

> In another published study, form the same journal, pooling strategy was again used to test thousands of genes (13528) and only few were validated, even finding some discrepancy between arrays and PCR (e.g. TMEM167A gene expression which was regulated in an opposite direction between array and qPCR). This is accepted even if the study did not include any other validation (Liamin et al., 2018).

> Pools have been used also in other animal models characterization (eg. pig) in terms of gene expression and miRNA (Martini et al., 2013).

Ref:

Łuczkowska et al. Molecular Mechanisms of Bortezomib Action: Novel Evidence for the miRNA-mRNA Interaction Involvement. Int J Mol Sci, 2020; 21: 350.

Liamin et al., Genome-Wide Transcriptional and Functional Analysis of Human T Lymphocytes Treated with Benzo[a]pyrene. Int J Mol Sci, 2018; 19: 3626.

Martini et al. Systems Biology Approach to the Dissection of the Complexity of Regulatory Networks in the S. scrofa Cardiocirculatory System. Int J Mol Sci, 2013; 14: 23160.

To sum up, due to my concerns on the validity of the array data and because authors did not provided any validation data, I cannot recommend the acceptance of the paper.

As already stated, we are aware that this Reviewer is ideologically against the use of pooling in the arrays strategy and he/she will not accept any arguments or objective proof about the quality of the assay.